# Applications of Surface Plasmon Resonance and Biolayer Interferometry for Virus–Ligand Binding

**DOI:** 10.3390/v14040717

**Published:** 2022-03-29

**Authors:** Shruthi Murali, Richard R. Rustandi, Xiwei Zheng, Anne Payne, Liang Shang

**Affiliations:** Analytical Research & Development, Merck & Co., Inc., West Point, PA 19486, USA; shruthi.murali@hotmail.com (S.M.); richard_rustandi@merck.com (R.R.R.); xiwei.zheng@merck.com (X.Z.); anne_payne2@merck.com (A.P.)

**Keywords:** surface plasmon resonance (SPR), biolayer interferometry (BLI), virus, virus-like particle (VLP), binding characterization, diagnostics

## Abstract

Surface plasmon resonance and biolayer interferometry are two common real-time and label-free assays that quantify binding events by providing kinetic parameters. There is increased interest in using these techniques to characterize whole virus-ligand interactions, as the methods allow for more accurate characterization than that of a viral subunit-ligand interaction. This review aims to summarize and evaluate the uses of these technologies specifically in virus–ligand and virus-like particle–ligand binding cases to guide the field towards studies that apply these robust methods for whole virus-based studies.

## 1. Introduction

### 1.1. History and Significance

Throughout the history of biomolecule binding detection methods, a common limitation has been the requirement for labeling molecules with reporter tags, which may change the conformation of the biomolecule or be a steric hindrance [1]. Surface plasmon resonance (SPR) mitigates this issue as a label-free, real-time technique that takes advantage of the change in refractive index of a thin metal sheet to provide information on the binding kinetics of an interaction [1]. Early exploration of SPR was conducted as studies on optical excitation of surface plasmons on smooth surfaces [2], which allowed for SPR to be further developed into an acceptable biosensing method [3]. The development of SPR is being followed by an adapted technology, biolayer interferometry (BLI), which employs the same principles of label-free and real-time binding but uses a simpler apparatus characterized by a biosensor that is dipped in analyte solution, making it more high-throughput than SPR.

To date, SPR and BLI have been used to detect binding between a wide variety of molecules, including protein–protein [4,5], protein–ligand [6,7], protein–DNA [8], etc. Currently, a large part of the literature on SPR or BLI applications are regarding biomolecule–biomolecule binding, whereas virus–biomolecule binding cases are not as prevalent. It is important to use these methods for virions or virus-like particles because they most closely mimic virus–receptor interactions during infection and share conformational similarities in exposed epitopes that may have roles in pathogenesis. Quantifying virus-biomolecule binding can provide significant information on the structure and function of viral antigens that may be used in vaccine development, or in predictive immunity studies for variable viruses, such as SARS-CoV-2 or influenza. Thus, this review aims to broadly expand on the principles, current applications, and limitations of SPR and BLI specifically for virus–ligand and virus like particle–ligand binding characterization.

### 1.2. Basic Principles of Surface Plasmon Resonance

SPR methods are characterized by three main components: the immobilized recognition molecule, the analyte molecule, and the prism of light. Figure 1 shows the mechanism for a typical SPR experiment. The immobilized molecule is bound to the sensor chip surface, which has a thin layer of metal, typically gold or silver. The analyte molecule is prepared in buffer and flows across the surface of the sensor chip, allowing the analyte molecule to bind the immobilized molecule. Prior to the binding event, the incident light wave passing through the prism excites the electrons in the metal film, forming surface plasmons. As the incident light is introduced at various angles, there is a critical angle where the photons are absorbed by the induced plasmon wave, a parameter affected by the refractive index of the medium. The critical angle for the immobilized molecule is shifted after the binding event, as the change in mass of the immobilized layer changes the refractive index of the sensing medium near the sensor chip surface [9,10,11,12].

There exists several parameters in experimental design, including the type of laser, type of metal, light wavelength, geometry of the prism, and methods of molecule immobilization. Biacore (Cytiva, Marlborough, MA, USA) is the widely used SPR biosensing system, and it helps to answer questions on interaction specificity, affinity, binding kinetics, and binding thermodynamics [9].

**Figure 1 viruses-14-00717-f001:**
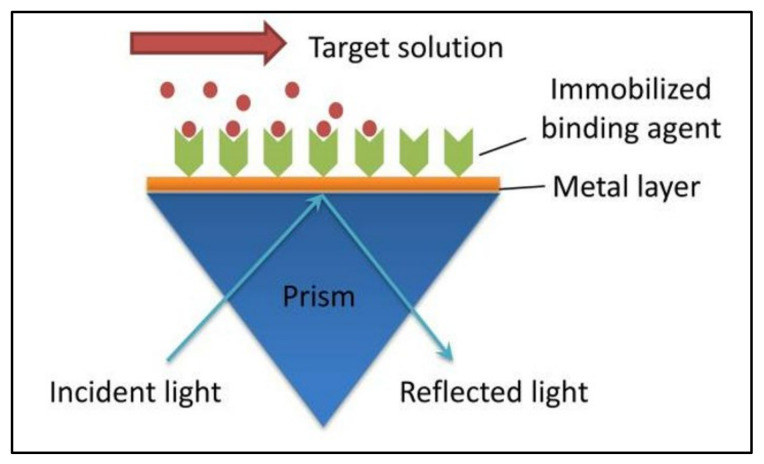
Surface plasmon resonance (SPR) mechanism. Reprinted from the *Journal of Pharmaceutical and Biomedical Analysis*, Vol. 113, Zheng, X., Bi, C., Li, Z., Podariu, M., Hage, D.S. [10], Analytical methods for kinetic studies of biological interactions: A Review, p. 168, (2015), with permission from Elsevier.

### 1.3. Basic Principles of Biolayer Interferometry

BLI systems are characterized by disposable biosensors that have a layer of the recognition molecule immobilized on the sensor tip, as Figure 2 shows [13,14,15]. The biosensor tip can then be dipped into wells containing the analyte solution, where binding of the analyte molecule to the recognition molecule is measured as disturbances to the interference pattern of white light that is directed down the length of the biosensor. The light is reflected off two surfaces—a ligand-binding surface and a stationary surface—where the binding event causes a shift in the wavelength of light that is proportional to the optical depth of the binding surface, which is detected and reported [14,15]. BLI is advantageous in comparison to SPR in its simplicity of dipping biosensors as opposed to using flow channels. This in turn makes the apparatus less expensive when compared to the SPR setup, with the added advantage of higher throughput potential [14].

ForteBio Octet (Menlo Park, CA, USA) is a commonly used BLI platform, and provides similar quantitative information as SPR systems, including binding kinetics, affinity, and specificity [14,15]. Much like SPR, key considerations in BLI experimental design include the selection of ligand to be immobilized and the immobilization method. Additionally, optimization may be necessary for each protocol and molecule pair, as deviations in results could manifest as issues in ligand saturation of the biosensor or nonspecific binding events [14].

### 1.4. Data Output and Interpretation of SPR and BLI Experiments

The output generated from SPR and BLI experiments include a response curve (sensorgram) and kinetic parameters that describe the binding events, including association, dissociation, and equilibrium constants. A theoretical sensorgram for SPR and BLI experiments is shown in Figure 3, and an experimental sensorgram generated by Chenail et al. [16] using SPR is shown in Figure 4. As the immobilized ligand binds the analyte molecule, the signal intensity increases in SPR due to a change in the refractive index of the medium on the metal surface, and in BLI due to a change in the optical depth of bound molecules on the tip surface. This portion of the curve can reveal the kinetics of association. This signal will continue to increase until a steady-state is achieved, indicating that all the recognition immobilized molecule binding sites are in an equilibrium of saturation with the target analyte. This portion of the curve may be used to deduce information about the binding affinity and equilibrium constant of the binding interaction. The signal will decrease as target molecules release from the ligand binding surface, from which the dissociation constant may be determined. At the end of dissociation, the sensor chip may be regenerated with stripping buffers to remove any remaining bound molecules to prepare the surface for additional experiments [10,14,17,18].

### 1.5. Comparable Binding Assays and Advantages and Limitations of SPR and BLI

SPR and BLI have many advantages over comparable binding assays, including enzyme-linked immunosorbent assay (ELISA), isothermal titration calorimetry (ITC), and flow cytometry. BLI and SPR are real-time, unlike ELISA and flow cytometry. Both technologies are also label-free, an advantage over the requirement of fluorescent labels for flow cytometry. The protocols for both platforms have been well-developed and are highly replicable, less prone to human error, real-time, and label-free, making them preferable to traditional assays in characterizing molecular interactions [15,19]. While ITC is label-free and real-time, its mechanism of characterizing binding interactions through enthalpy changes in solution results in a method more sensitive to matrix changes than SPR and BLI. This makes it less robust, as nonspecific enthalpic changes resulting from events such as matrix dilution may be dominant in the total enthalpy output, making it difficult to distinguish the enthalpic changes resulting from the binding partners [20]. Beyond these conventional binding assays, there are several electrochemical platforms offering real-time, sensitive detection [21].

The primary limitation of SPR and BLI is that one binding partner is immobilized on a surface, which may affect the binding event [22]. It is thus necessary to ensure that the immobilization method does not alter the conformation or orientation of the biomolecule. SPR and BLI are also not suitable for small target molecules, as their mass may not allow for a distinguishable critical angle shift [23]. Additionally, it is crucial that regeneration of the sensor chip surfaces be with an optimized buffer that does not structurally affect the immobilized molecules [16,24].

An informative comparison to these alternative binding assays with advantages and limitations is shown in Table 1.

The following sections will highlight SPR and BLI application case studies for the measurement of virus–biomolecule binding with an aim to showcase the variety of configurations that can be implemented and the breadth of output information that can be obtained from these systems.

## 2. SPR Case Studies

### 2.1. SPR for Virus–Ligand Binding

Several studies have applied SPR to develop detection or diagnostic platforms for viruses using different ligands, such as aptamers and antibodies. In this section, we broadly cover the applications of SPR to aptamer-based detection of avian influenza [28,29,30], the improvements to existing immobilization methods with the model viruses mumps [31], cytomegalovirus [16], and oyster mushroom spherical virus [32], and plant [33,34] and animal virus [35,36,37] detection and characterization.

The key contents of the SPR virus–ligand case studies are summarized in Table 2.

#### 2.1.1. Avian Influenza Virus

A range of studies on avian influenza have been conducted with a focus on detecting whole H5Nx viruses with aptamers using SPR. The platform proves useful in distinguishing aptamers with high affinity to whole virus, with applications in diagnostic development. One method for viral detection shown in two independent studies involved the use of a pair of aptamers to detect the whole virus in a sandwich format, where one biotin-tagged aptamer was immobilized to the sensor chip surface with streptavidin [28,29]. Given that a sandwich format requires multiple interactions to confirm binding, it provides the advantage of greater certainty in viral detection. Using SPR as the detection platform, Nguyen et al. [28] demonstrated specific detection of H5N1 by the advantage of having highly specific aptamers to two different regions of the virus.

In another aptamer sandwich-based SPR system, Kim et al. [29] developed a lateral flow strip assay for H5N2, using SPR to screen for the highest affinity aptamer pair among those tested. While SPR was not the detection platform itself in this case, the method provided valuable characterization of screened aptamers to allow the development of a highly specific and sensitive detection platform with readout specific to H5N2 in comparison to other H5Nx viruses.

A contrasting study developed a portable SPR biosensor to detect H5N1 using the Spreeta SPR sensing chip (Texas Instruments, Dallas, TX, USA), which contains the general SPR apparatus in a compact size. This sensor makes use of a single immobilized aptamer but demonstrated success by detecting H5N1 in poultry swab samples. The authors discuss the benefits of using SPR over PCR for detection methods, namely the inability of PCR to distinguish between live and inactivated viruses [30]. SPR is a strong method to ensure specific, sensitive, and rapid detection, and this case is indicative of the capabilities of portable SPR biosensors.

#### 2.1.2. Mumps Virus

The wide variety of experimental configurations makes SPR a versatile method with opportunities for optimization. For example, a common method of immobilizing virions on a sensor chip surface is via a cationic polymer such as poly-l-lysine. The polymer is adsorbed onto an 11-mercaptoundecanoic acid (MUA) monolayer, which is prepared on a gold array. While this is a widely used method, Kim et al. [31] raise the perspective of matrix pH altering polymer charge, which in turn reduces the efficiency of virus capture. In their study aiming to develop a detection method for mumps virus, an alternative surface was prepared by replacing poly-l-lysine with the positively charged polyelectrolyte, poly(diallyldimethylammonium chloride) (PDDA). This substitution demonstrated increased virus binding capacity in comparison to the poly-l-lysine surface, an improvement that could be attributed to PDDA’s greater resilience to matrix conditions [31].

#### 2.1.3. Cytomegalovirus

A key component of virus–ligand SPR experiments is the method of immobilization. In many cases, a virus is immobilized by attaching to a capture antibody or receptor, where the sensor chip regeneration step removes all bound viruses and new viruses are bound for subsequent runs. An alternative approach was taken in a cytomegalovirus (CMV) experiment to quantify binding kinetics and affinity of antibodies to whole virions, where whole virions were immobilized directly onto the sensor surface. Although there are concerns of conformational change during sensor chip regeneration when the virus is directly immobilized, Chenail et al. demonstrated that conditioning experiments can be used in coordination with microscopy to determine an amenable regeneration buffer that does not perturb virion conformation. By testing various regeneration buffers at different concentrations and pH, they determined that 25 mM NaOH at a flow rate of 100 μL/min for one minute was the superior buffer in this case for removing bound antibody and retaining virion structure and binding ability. Another key aspect of this work was the difference between equilibrium binding constants found through SPR and ELISA. This indicates that while it is possible to determine the binding affinity (K_D_) from ELISA, it may not be as accurate as that determined from a more sensitive platform with the ability to estimate kinetic binding constants, emphasizing the greater reliability of SPR [16].

#### 2.1.4. Plant and Fungal Viruses

One of the earlier studies using SPR to characterize binding interactions of whole virus was done in 2004, where Boltovets et al. [33] designed a method to detect tobacco mosaic virus (TMV). Given its small size, TMV would serve as a model to guide future studies that may attempt to characterize larger viruses, especially because of the molecular size limitation in SPR that may prevent accurate quantitation if surpassed. A key technique in this study was the immobilization of protein A to the sensor surface, and the subsequent binding of pre-incubated antibody–virus complexes. This would increase the compactness of bound antibody–virus complexes to the protein A sensor surface in comparison to a virus detection event using only a single antibody. As this is recorded as a higher density per area event, it results in an increased critical angle shift [33].

The majority of SPR technique characterization studies occurred during the onset of its wide usage, thus allowing it to be a supporting assay to provide information that would otherwise be assumed from tangential assays. One example is in its use to characterize the binding kinetics of antibodies to the veinal necrosis strain of potato virus Y (PVY) to guide the development of a sandwich lateral flow immunoassay (LFIA). Data from SPR technology demonstrated the presence of antibody–PVY complexes in the LFIA [34].

The application of SPR to fungal virus detection applies the same techniques, but discusses the advantages of a dextran-based immobilization method. Anti-OMSV (oyster mushroom spherical virus) mAbs were obtained by immunizing purified virus in mice and collecting sera. These mAbs were immobilized onto a dextran layer on the biosensor chip surface. Dextran was used in this application as it contains more mAb binding sites than protein A-based immobilization, with the caveat of protein A conserving uniform orientation. In this application, Kim et al. [32] compare the more pronounced SPR angle shift from their dextran-based sensor to a smaller shift seen in previous studies using protein A to support their conclusion that using dextran is an improved strategy [32]. However, a comparison is made to SPR experiments with other viral species. Thus, further studies with OMSV and protein A would fully elucidate if one immobilization method is more favorable.

#### 2.1.5. Animal Viruses

The application for SPR on animal viruses follows plant-based applications, of which an early study in 2006 used the model insect pathogen *Autographa californica* multiple nuclear polyhedrosis virus (AcMNPV). The mAb AcV1, raised against AcMNPV surface protein gp64, was immobilized to the gold sensor chip using Protein A, where the virion served as the target analyte [35].

SPR has since been applied to several other animal models. It has been shown to be highly useful in determining assay modifications that can reduce the limit of detection (LOD). For example, in the detection of bovine viral diarrhea virus type 1 (BVDV type 1), the LODs of three configurations (single immobilized capture aptamer, sandwich aptamer capture, and sandwich aptamer capture where one aptamer contains a gold nanoparticle (AuNP) label) were compared. The results indicated that the highest LOD was with the single aptamer, and the lowest with the AuNP-labeled sandwich pair. The authors discuss the wide usage of AuNPs in sandwich configurations for a stronger readout, which may be the reason for PCR-comparable sensitivity in the AuNP sandwich group. SPR excels in this scenario to provide a real-time quantitative comparison between the three different detection mechanisms [36].

The application of SPR to develop a detection method with a lower LOD has been replicated for other viruses where the levels required to cause illness are significantly lower than what is commonly detected. For example, an assay was developed to detect feline calicivirus, a surrogate for norovirus, using immobilized anti-FCV antibodies to capture the virus and secondary antibodies to detect the virus presence [38]. Another example in is in the development of a diagnostic for porcine reproductive and respiratory syndrome virus (PRRSV) strain VR-2332, where SPR was used to find an aptamer with high specificity towards PRRSV to be used in the development of a sandwich aptamer-based detection platform [37].

### 2.2. SPR for Virus-Like-Particle (VLP)–Ligand Binding

In this section, we discuss the application of SPR to VLPs specifically for epitope mapping to characterize simultaneous binding events [24] and the characterization of norovirus antibodies [39].

The key contents of the SPR VLP–ligand case studies are summarized in Table 3.

#### 2.2.1. Viral Epitope Mapping

A benefit of SPR in comparison to ELISA is its ability to clearly explain simultaneous binding events on the same molecule. Where in ELISA the binding of different analytes to a single recognition molecule cannot be distinguished as only a holistic binding signal is presented, SPR allows the constant monitoring of multiple binding and disassociation events. This makes SPR a useful tool to map epitopes on antigens and determine which analytes competitively bind the same site and which ones can bind simultaneously. This assay was applied by Towne et al., to assess eight monoclonal antibodies (mAbs) against human papillomavirus type 16 (HPV 16) VLP. By immobilizing an anti-HPV antibody to capture the VLP, pairs of mAbs were introduced to assess binding sites. The data establishes that association curves occur in succession during successful simultaneous binding events, but combined offloading and onloading is seen when mAbs compete for the same site [24].

#### 2.2.2. Norovirus VLP and Antibody Characterization

ELISA may also be used in conjunction with SPR to provide additional information, specifically on the reactivity of masked epitopes. In a study characterizing the capture of norovirus (NoV) mAbs using NoV VLPs from 16 different NoV genotypes, SPR was used to find the dissociation constant and ELISA was used to confirm the successful binders. Although ELISA relies on binding measured through surface adsorption and this may include non-specific binders, the authors propose its advantages in discovering masked epitopes due to conformational changes that occur as the capture molecule is immobilized on the plate surface [39].

## 3. BLI Case Studies

### 3.1. BLI for Virus–Ligand Binding

A large number of case studies using BLI for virus–ligand interaction are concerned with structural studies of the influenza A virion or the characterization of flu virus-receptor binding. The latter would address the virus’ zoonotic potential to evolve from binding the avian receptor analogue towards binding the human receptor analogue. Here, we broadly summarize the key studies on how BLI has been applied to study the surface glycoprotein balance on the flu virion [40,41,42] and the evolution of binding preferences of the strains H3N2 [43], H5N1 [44,45], and H9N2 [46,47]. Two additional cases are summarized on parvovirus capsid binding to receptors and antibodies [48] and the characterization of a tripartite complex involving dengue virus [49].

The contents of the BLI virus–ligand case studies are summarized in Table 4.

#### 3.1.1. Structural Studies of Influenza A Virus

The focus in structural flu virion studies is the balance of hemagglutinin (HA) and neuraminidase (NA) on the virion surface, as HA binds the sialic acid host receptor to promote infection, while NA cleaves the receptor. Because a balance between these two antigens is crucial for viral replication, structural characterization should highlight the relative contributions of each antigen. An approach was developed for this question using BLI in 2015, showing the binding of virus to human and avian receptors in the presence and absence of NA inhibitors. In the presence of NA inhibitors, the activity of HA can be deduced, as binding is favorable. In the absence of NA inhibitors, sugar depletion assays revealed the receptor cleaving activity of NA. BLI sensorgrams from both experiments can be used to develop a ratio that describes the HA/NA balance [40]. A subsequent study focused on a second sialic acid binding site that NAs of avian viruses contain and used BLI to characterize the site, showing that it should be considered an additional parameter in HA/NA balance BLI experiments, as the presence of such a site affects the binding and cleaving balance of both glycoproteins [41].

In another structural influenza binding study, Guo et al. expanded on the previous uses of BLI to report K_D_ values for flu virion and host cell receptor binding. However, they challenge these conclusions with their BLI experiments that demonstrate an irreversible binding event with an off-rate constant of zero, and attribute this to the multivalent interactions between the virion and receptors. The advantage of using the BLI sensorgram in this case is the ability to determine the initial binding rate, which may not be possible with endpoint assays [42].

#### 3.1.2. Binding Preferences of H3N2

The H3N2 virus was responsible for the 1968 flu pandemic and was caused primarily by the evolution in binding of hemagglutinin, a flu surface glycoprotein, to human receptors from avian receptors. Over the course of the last ten years, binding has decreased between the virus and human receptor analogs. This viral evolution was studied by Lin et al. [43] using BLI to quantify viral interactions to avian and human sialic acid receptor analogs (α2,3-linked sialic acid and α2,6-linked sialic acid, respectively). Receptor analogs were immobilized to the sensor surface, and whole H3N2 virus particles were flowed through the analyte buffer. They determined how much various amino acid mutations in hemagglutinin reduce the virus–receptor interaction, and compared these results to BLI experiments using the viruses in circulation during the evolutionary period. This allowed the study to conclude which amino acid substitution was a primary contributor to the decreased affinity seen in the 2005 virus [43].

#### 3.1.3. Binding Preferences of H5N1

Similar studies have been conducted for H5N1, specifically to screen receptor binding preferences of whole recombinant viruses [44] and to quantitate virus binding in Southeast Asian clade I and in mutant clade 2.2 viruses endemic to Egypt [45].

In studying recombinant viruses, the agglutination of horse and guinea pig red blood cells (RBCs) by various human and avian flu strains was measured. Given that horse RBCs contain the avian 2,3-sialyl sequences and guinea pig RBCs contain the avian 2,3- and human 2,6-sialyl sequences, this assay would indicate which strains bound only the avian receptor and which bound both avian and human receptors. Using eleven H5N1 viruses isolated from humans, they found that three constructs showed lower or no binding to horse RBCs, indicating a reduced preference for the 2,3-sialyl sequence. By creating recombinant viruses with two mutations that mimicked the human isolate binding preferences, BLI experiments were conducted to determine by how much the mutations reduced the affinity of the recombinant virus to the avian receptor, and to observe a binding increase to human receptors [44]. BLI in this scenario is a useful technology to follow up on the preliminary agglutination data, as it reduces the sample size.

The studies on this topic continue to follow a similar pattern of using BLI to comparatively quantify receptor binding to mutant flu viruses. In a study by Xiong et al., a key finding was the distinction that the introduction of double substitutions in recombinant viruses can have opposing effects on binding, i.e., one substitution may weaken binding, but a second may restore binding [45]. BLI is highly useful in these scenarios to provide a platform where the cause for such effects can be immediately recognized.

#### 3.1.4. Binding Preferences of H9N2

Peacock et al. have studied receptor-binding avidity of H9N2 using BLI in two studies [46,47], as the zoonotic potential of H9N2 viruses had not yet been characterized. In 2017, the authors tested several H9N2 isolates to determine that a majority preferred to bind the sulfated avian-like receptor analogues over the non-sulfated receptor. By analyzing the K_D_ values of different isolates to the human, avian, and sulfated avian receptors, this study provides a basis of particular isolates that pose a zoonotic threat [46]. In 2020, the authors chose three of the isolates characterized in 2017 to investigate effects of mutagenesis on receptor-binding residues. The viruses chosen included one with a preference for only the sulfated avian receptor, one that would bind both sulfated and non-sulfated avian receptor, and one that would bind all three receptors but demonstrated a preference for the human receptor. By introducing mutations into the hemagglutinin present on each of these viruses, BLI was used to explain the differences in receptor-binding preference between the three isolates [47].

#### 3.1.5. Other BLI Applications for Virus–Ligand Binding

Beyond structural characterization and mutagenesis experiments, BLI is valuable for competition-based assays. Callaway et al. [48] characterized the binding interaction between parvovirus capsids and their host cell receptor, transferrin receptor type 1 (TfR) of various species, and the binding competition of fragment antibodies (Fabs) to capsids bound to TfR and scFv-Fcs. Key components to this BLI application include determination of the number of receptors bound to TfR through antibody competition assays by assuming a saturating number of Fabs for Fab–capsid interactions and comparing the level of TfR binding by normalizing to the masses of both molecules [48].

BLI has also been applied to study interactions that exacerbate infections, extending beyond its scope to characterize mechanisms of infection and neutralization. For example, fibrinolysis is a symptom of Dengue, characterized by fibrin clot degradation by plasmin. The binding of plasmin to DENV is studied using BLI to determine if the interaction enhances infection, and a relevant inhibitor AaTI is tested to gauge the suppression of infectivity. BLI was used to first quantify plasmin–AaTI binding where AaTI is immobilized on the sensor surface and is dipped in plasmin solution. To assess DENV binding to the plasmin-AaTI complex, the biosensor containing plasmin-AaTI is dipped into a DENV solution. These experiments, supported by plaque assays from infected mosquitos, allow the conclusion that DENV infection is enhanced as plasmin concentrations increase, and can be inhibited through simultaneous binding of the AaTI inhibitor [49].

### 3.2. BLI for VLP–Ligand Binding

Apart from whole viruses, virus-like particles are of increasing interest in the development of vaccines and therapeutics because they are non-infectious, multivalent and highly immunogenic. They have been clinically demonstrated to elicit robust immune responses, and studies have shown they may be more potent than inactivated viruses [50]. Thus, studying VLP interactions using BLI will provide a valuable basis of knowledge to promote the assay in clinical candidate development.

In this section, we discuss the usage of BLI to develop an antibody avidity assay through immunization with a DENV VLP vaccine [51], to characterize a heparin-like VLP and its anticoagulation activity [52], to screen and select tumor-specific antibodies by displaying HER2 on VLPs [53], and to explore modifications to enhance the stability of chimeric hepatitis B virus core antigen VLPs [54].

The contents of the BLI VLP–ligand case studies are summarized in Table 5.

#### 3.2.1. Dengue Virus (DENV) VLP Vaccine Antibody Avidity Assay

One key aspect of a robust immune response to a vaccine is antibody affinity maturation, which can be difficult to quantify. The phase 2 trials of the tetravalent DENV VLP vaccine developed by Takeda consisted of a novel BLI assay used to measure the avidity of serum antibodies to DENV antigens of serotypes 1–4. By immobilizing the DENV VLPs to the sensor tip and associating purified sera from different time points, the BLI curves demonstrate an increased affinity and decreased binding off-rate over time [51]. This study demonstrates the ease of comparing BLI experiments over time points to find real-time trends in the neutralization mechanism.

#### 3.2.2. Other VLPs: Anticoagulation Activity of Heparin-Like VLPs and Display of HER2 on HIV VLPs

Beyond virions and VLPs that mimic viruses, a wide variety of antigens can be displayed on VLPs to take advantage of the polyvalency. One example is the sulfated bacteriophage Q-β (sulf-VLP), a virus-like nanoparticle that exhibits anticoagulation activity like heparin. Because the purification process for heparin is extensive, this alternative may be clinically used if the interaction is comparable to that of heparin. BLI was used to characterize the binding on and off events of sulf-VLP to a cationic peptide, CDK5 [52]. Another example is the display of full-length HER2 on the surface of HIV-1 Gag-derived VLPs. This study aimed to recommend a promising scaffold to display membrane receptor protein targets as immunogens to identify new therapeutic antibodies. BLI was applied to test the functionality of displayed HER2 in binding the anti-HER2 antibody [53]. One limitation of working with VLPs is in the dissociation range, where weak dissociation reinforces the VLP multivalency, which may not mimic the natural off-rate of the antigen in its native presentation.

#### 3.2.3. Stability of Chimeric Hepatitis B Virus Core Antigen VLPs

While BLI does not provide a platform to gauge physical or thermal stability of molecules, the mechanisms that result in stability can be deduced. It was demonstrated by Schumacher et al. that a C-terminal histidine-peptide (His6-peptide) addition to chimeric hepatitis B virus core antigen VLPs stabilizes in chemical and physical stress tests [54]. To determine the mechanism of stability, the intra- and inter-subunit interactions of the VLP were determined using BLI for disassembled and intact VLPs. The two VLPs were immobilized on the sensor surface and interaction of the anti-His6-peptide antibody was measured. This experiment revealed that the anti-His6-peptide antibody only binds to the dissembled VLP, indicating the stabilizing peptide is only accessible in an internal, unexposed epitope. While BLI is not at the core of this study, it provides valuable support to understand the interaction.

## 4. Conclusions

### 4.1. Concluding Remarks on Applications of SPR and BLI

SPR and BLI have both evolved to provide versatile, sensitive, specific, and rapid assays for the detection of a wide variety of viruses paired to a range of ligands and receptors. This review has focused on the advances for viral characterization in the contexts of infection and neutralization, diagnostics, epitope mapping, particle stability, and method optimization. The flexibility and replicability of these platforms make them favorable candidates to characterize binding events thoroughly, and elucidate virus and VLP properties with greater confidence than with traditional binding assays.

### 4.2. Future Directions

There are a few main limitations to both of these technologies—namely the extensive optimization required for sensor chip regeneration, selecting an immobilization method that does not interfere with the binding event, controlling for non-specific binding events and ensuring optimal binding conditions for the target interaction, and mass transport limitations [55], specifically in the SPR apparatus. However, the advantages may give rise to interest in the development of portable SPR and BLI platforms for accurate point-of-care diagnostics [56], as methods such as the miniature SPR Spreeta [30] discussed in this review make use of the convenient apparatus for viral detection.

We believe the future of SPR and BLI may hold a platform that supports binding characterization of larger particles, including cells, and advancement of sensor chips that limit non-specific interactions and promote proper conformation of the target immobilization molecule. Additionally, the large expenses associated with using these instruments and their sensor chips could be reduced by exploring sensor chip regeneration for reuse.

## Figures and Tables

**Figure 2 viruses-14-00717-f002:**
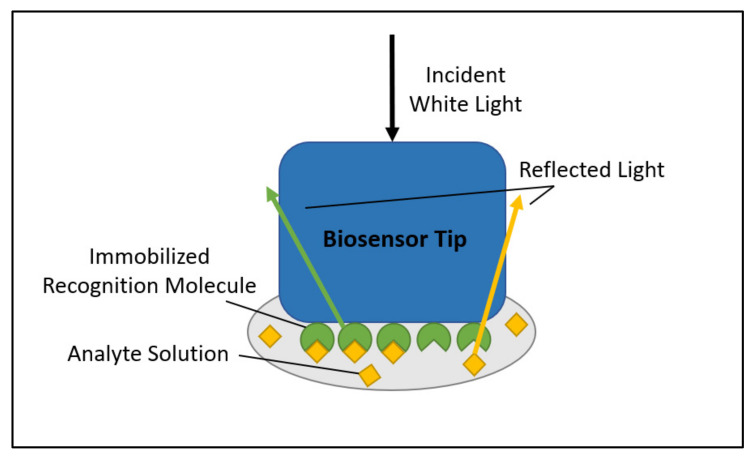
Biolayer interferometry (BLI) mechanism.

**Figure 3 viruses-14-00717-f003:**
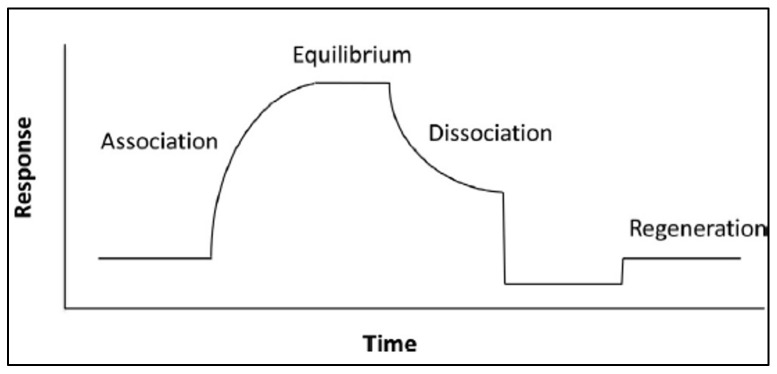
Theoretical sensorgram. Reprinted from the *Journal of Pharmaceutical and Biomedical Analysis*, Vol. 113, Zheng, X., Bi, C., Li, Z., Podariu, M., Hage, D.S. [10], Analytical methods for kinetic studies of biological interactions: A Review, p. 168, (2015), with permission from Elsevier.

**Figure 4 viruses-14-00717-f004:**
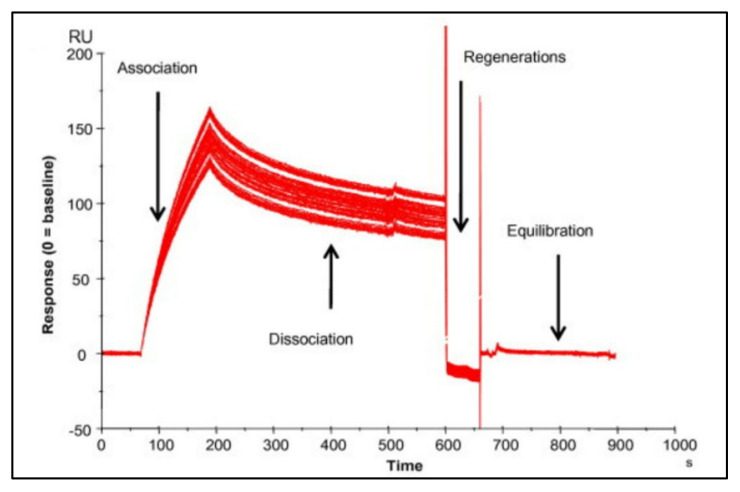
Experimental SPR sensorgram. Reprinted from *Analytical Biochemistry* Vol. 411/Issue 1, Chenail, G., Brown, N.E., Shea, A., Feire, A.L., Deng, G. [16], Real-time analysis of antibody interactions with whole enveloped human cytomegalovirus using surface plasmon resonance, p. 60, (2011), with permission from Elsevier.

**Table 1 viruses-14-00717-t001:** Comparison of surface plasmon resonance (SPR) and biolayer interferometry (BLI) to other common binding assays.

Assay	Method Overview	Advantages	Limitations
SPR [19]	Binding measured with one immobilized partner and one partner in flow chamber by reflection of light.	Label-free, real-time kinetic	Regeneration optimization [16,24] Non-specific binding One immobilized partner
BLI	Binding measured with one partner immobilized and one diluted partner in plate well by reflection of light.	Label-free, real-time kinetic	Sample evaporation over time [25] Agitation required to prevent re-binding [25] One immobilized partner Regeneration optimization Non-specific binding
Enzyme-linked immunosorbent assay (ELISA)	Binding measured as surface adsorption between both binding partners by chemical colorimetric signal [19].	Label-free	Not real-time Several buffers Plate-based immobilization Signal based on secondary antibody [19] Non-specific binding
Isothermal Titration Calorimetry (ITC) [20]	Binding measured in solution by enthalpy changes from binding event.	Label-free, real-time kinetic	High sensitivity to matrix changes and non-specific enthalpic events
Flow Cytometry [26,27]	Binding measured between two suspended partners using fluorescence and light scattering.	Supports cell-based assays	Not real-time Fluorescent labeling required

**Table 2 viruses-14-00717-t002:** Summary of SPR virus–ligand cases discussed in Section 2.1.

Virus	Ligand	Immobilization	Matrix	Key Findings	Platform	Reference
Cytomegalovirus (CMV)	CMV glycoprotein B anti-antibodies	CM3 sensor chip, amine coupling of CMV virons	PBS, 0.05% Tween-20	Table 1 of the study contains binding affinity and kinetics values.	Biacore T100	Chenail et al., 2011 [16]
H5N1, H5N8, and H5N2 viruses	Biotin-labeled H5Nx-specific aptamers	Streptavidin biosensor, biotinylated aptamers	N/A	SPR-based biosensor for H5N1 had a limit of detection (LOD) of 200 EID_50_/mL in detecting virus in infected feces samples.	Eco Chemie, Utrecht, The Netherlands	Nguyen et al., 2016 [28]
H5N2 virus	Aptamers	Streptavidin biosensor, biotinylated aptamers	N/A	Screened for the aptamer pair with the most intense signal compared to other pairs.	N/A	Kim et al., 2019 [29]
H5N1 virus	Aptamer	Streptavidin biosensor, biotinylated aptamer	PBS	Best aptamer had a dissociation constant of 4.65 nM.	Miniature Spreeta SPR detector	Bai et al., 2012 [30]
Mumps virus	Mumps-specific mAb	Poly(diallyldimethylammonium chloride) (PDDA) to immobilize mumps virus	Virus incubation: PBS, 3.7% formaldehyde Washes in between Ab loading: PBS, 0.1% Tween-20	Demonstration of increased surface adsorption by immobilizing using PDDA instead of with poly-L-lysine.	Non-commercial, self-developed	Kim et al., 2006 [31]
Oyster mushroom spherical virus (OMSV)	Anti-OMSV mAb from mice sera	Activated carboxymethyl-dextran sensor, mAb immobilized	10 mM HEPES pH 7.4	Dextran-based mAb immobilization gives a stronger SPR response than Protein A-based immobilization seen in previous studies.	AutoLab ESPRIT (Utrecht, The Netherlands)	Kim et al., 2008 [32]
Tobacco mosaic virus, strain TMV-U	TMV-specific IgGs	Protein A, Fc region of IgGs	Glycine buffer, pH 2.2 Carbonate buffer, pH 9.6	Pre-incubation of virus with IgG results in increased critical angle shift during SPR.	PLASMON BioSuplar (New York, NY, USA)	Boltovets et al., 2004 [33]
Potato virus Y (PVY)	Anti-PVY^N^ antibodies	CM5 sensor chip, amine coupling of mAbs	10 mM HEPES pH 7.4 150 mM NaCl 0.005% Tween-20	k_d_ ^1^ = (3.9 ± 0.4) × 10^−4^ s^−1^ k_a_ ^2^ = (2.9 ± 0.5) × 10^4^ M^−1^·s^−1^ K_D_ ^3^ = 1.4 × 10^−8^ M	Biacore X	Razo et al., 2018 [34]
Austographa californica multiple nuclear polyhedrosis virus (AcMNPV)	Monoclonal antibody (mAb) AcV1 against gp64, a surface protein	Amine reactive crosslinker for DTSSP SAM + Protein A + IgG	50 mM phosphate buffer	The platform used was sensitive to AcMNPV and not to the control virus, promoting SPR use for human and animal samples.	Non-commercial, self-developed	Baac et al., 2006 [35]
Bovine viral diarrhea virus (BVDV) type 1	Aptamers, for sandwich assays	Streptavidin biosensor, biotinylated aptamer	N/A	LOD goes from 10^4^ (single aptamer) to 5000 (sandwich aptamer) to 500 TCID_50_/mL (sandwich with gold nanoparticle).	Eco Chemie, (Utrecht, The Netherlands)	Park et al., 2014 [36]
Porcine reproductive and respiratory syndrome virus (PRRSV) type II, Strain VR-2332	Single-stranded aptamer	Streptavidin biosensor, biotinylated aptamer	N/A	Novel single-stranded aptamers specific to PRRSV type II, strain VR-2332 reported, with a binding affinity of 2.5 × 10^5^ TCID_50_/mL	Eco Chemie, (Utrecht, The Netherlands)	Lee et al., 2013 [37]
Feline calicivirus (FCV)	Anti-FCV Abs	CM3 dextran sensor chip, amine coupling of Abs	0.01 M HEPES pH 7.4 0.15 M NaCl	This sandwich biosensor detects FCV particles with a limit of detection of 10^4^ TCID_50_ FCV/mL	Biacore T100	Yakes et al., 2013 [38]

^1^ k_d_, dissociation rate constant; ^2^ k_a_, association rate constant; ^3^ K_D_, equilibrium constant.

**Table 3 viruses-14-00717-t003:** Summary of SPR VLP–ligand cases discussed in Section 2.2.

Virus	Ligand	Immobilization	Matrix	Key Findings	Platform	Reference
Human papillomavirus (HPV) type 16 virus-like particle (VLP)	8 HPV-16 VLP monoclonal antibodies (mAbs)	CM5 sensor chip, amine coupling of mAbs	10 mM HEPES 150 mM NaCl 3 mM EDTA 0.005% Tween-20 pH 7.4	All tested antibodies bind to related epitopes on HPV-16 VLPs.	Biacore 3000	Towne et al., 2013 [24]
Norovirus (NoV) VLPs	NoV monoclonal antibodies (mAbs)	CM5 sensor chip, amine coupling of rabbit anti-mouse IgG. Followed by mAbs.	0.01 M HEPES 0.15 M NaCl 0.05% Tween-20 pH 7.4	Table 3 in study displays all mAb-VLP dissociation constants k_d_.	Biacore 3000	Kou et al., 2015 [39]

**Table 4 viruses-14-00717-t004:** Summary of BLI virus–ligand cases discussed in Section 3.1.

Virus	Ligand	Immobilization	Matrix	Key Findings	Platform	Reference
Influenza A Virus, Strains X-31 and X-31 HAM	Human and avian receptor analogs: α2,6-sialyl-*N*-acetyllactosamine and α2,3-sialyl-*N*-acetyllactosamine	Streptavidin biosensor, biotinylated receptors	10 mM HEPES pH 7.4 150 mM NaCl 0.005% Tween-20 4 mM CaCl2	Novel assay to determine viral surface receptor balance based on virus binding to human and avian receptor analogs with and without neuraminidase inhibitors.	Octet RED (ForteBio, Menlo Park, CA, USA)	Benton et al., 2015 [40]
HK H3N2 hH3hN2, lacking the second binding site for NA and hH3aN2, containing the second binding site for NA viral strains. Avian NA contain a second binding site for sialic acid receptor.	2,3-sialyl-*N*-acetyllactosamine-*N*-acetyllactosamine and 2,6-sialyl-*N*-acetyllac- tosamine-*N*-acetyllactosamine and *N*-acetyllactosamine-*N*-acetyllactosamine and LAMP1 or glycophorin A glycoproteins	Streptavidin biosensor, biotinylated receptors	PBS with calcium and magnesium	The effects of the second binding site on NA should be considered in receptor balance studies.	Octet RED348	Du et al., 2019 [41]
Influenza A Virus, lab strains PR8_MtSIN_ and PR8_CAM2,3_	Sialic acid receptors and Fc-tagged glycoproteins	Streptavidin biosensor, biotinylated receptors and Protein A biosensor	PBS with calcium and magnesium	Binding of influenza A virus to ligand is irreversible when NA cleavage activity is insignificant. k_off_ ~ 0	Octet QK	Guo et al., 2018 [42]
H3N2	Human and avian receptor analogs: α2,6-sialyl lactosamine and α2,3-sialyl lactosamine	Streptavidin biosensor, biotinylated receptors	150 mM NaCl 10 mM Hepes pH 7.4 3 mM EDTA 100 μM oseltamivir carboxylate 0.005% Tween-20	Hemagglutinin Asp-225-Asn substitution is very critical in the decrease of human receptor binding from the 2004 to 2005 virus.	Octet RED	Lin et al., 2012 [43]
30 different H5N1 viruses	Human and avian receptor analogs: α2,6-sialyl lactosamine and α2,3-sialyl lactosamine	Streptavidin biosensor, biotinylated receptors	10 mM HEPES pH 7.4 150 mM NaCl 3 mM EDTA 0.005% Tween-20 Viral solutions incubated with 10 μM oseltamivir carboxylate and 10 μM zanamivir.	Analyzed three recombinant H5N1 viruses with BLI after agglutination studies. Mutations to residues 134 and 186 on HA result in weakened binding to human and avian receptor analogs.	Octet RED	Crusat et al., 2013 [44]
Recombinant H5N1	Human and avian receptor analogs: α2,6-linked sialic acid and α2,3-linked sialic acid	Streptavidin biosensors, biotinylated receptors	10 mM HEPES pH 7.4 150 mM NaCl 3 mM EDTA 0.005% Tween-20 10 μM oseltamivir carboxylate 10 μM zanamivir	Substitutions Asn186Lys and Ser227Asn in H5 clade 1 HA decrease affinity for avian receptor analog.	Octet RED	Xiong et al., 2014 [45]
H9N2–contemporary viruses of the zoonotic G1 lineage and representative viruses of the zoonotic BJ94 lineage	Human receptor analog: α2,6-sialyl lactosamine and Avian receptor analogs: α2,3-sialyl lactosamine, sulfated and non-sulfated	Streptavidin biosensors, biotinylated receptors	10 mM HEPES pH 7.4 150 mM NaCl 3 mM EDTA 0.005% Tween-20 10 μM oseltamivir carboxylate 10 μM zanamivir	K_D_ values presented in Table 1 of study.	Octet RED	Peacock et al., 2017 [46]
H9N2	Human receptor analog: α2,6-sialyl lactosamine and Avian receptor analogs: α2,3-sialyl lactosamine, sulfated and non-sulfated	Streptavidin biosensors, biotinylated receptors	10 mM HEPES pH 7.4 150 mM NaCl 3 mM EDTA 0.005% Tween-20 10 μM oseltamivir carboxylate 10 μM zanamivir	The effects of mutagenesis on HA receptor-binding residues explained by BLI. Residues 190, 226, and 227 have large effects on receptor binding preference.	Octet RED	Peacock et al., 2021 [47]
Canine and feline parvovirus capsid	Transferrin receptor type 1 and scFv-Fcs	Ni-NTA biosensors, His-receptors and Protein A biosensors, scFv-Fcs	PBS, 0.02% ovalbumin 0.02% Tween-20	Determined association constants, dissociation constants, and binding affinity between the capsid and receptor/antibodies.	BLItz (ForteBio)	Callaway et al., 2018 [48]
DENV	Kazal-type serine protease inhibitor, AaTI and Plasmin	Ni-NTA biosensors, His-AaTI	50 mM Tris pH 7.5 100 mM NaCl 0.1% BSA buffer	Plasmin bound rAaTI with K_d_ of 62.8 nM.	Octet RED 96	Ramesh et al., 2019 [49]

**Table 5 viruses-14-00717-t005:** Summary of BLI VLP–ligand cases discussed in Section 3.2.

Virus	Ligand	Immobilization	Matrix	Key Findings	Platform	Reference
DENV serotypes 1–4 VLPs	Anti-DENV Ab control Abs from vaccinated patient sera	Streptavidin biosensor, biotinylated VLPs	0.1% BSA-PBST	Avidity index (response/k_off_) increased by vaccination	Octet RED and Octet HTX	Tsuji et al., 2021 [51]
Sulfated Q-β bacteriophage, yields VLP-like particles	Cationic peptide CDK5 (50% Lys)	Streptavidin biosensor, biotinylated CDK5	HBS: 50 mM HEPES 100 mM NaCl pH 7.4 0.02% NaN_3_; 100 mM sodium phosphate pH 7 buffer	k_on_ = (8 ± 3) × 10^6^ s^−1^ k_off_ = (5 ± 2) × 10^−3^ Ms^−1^ Overall dissociation constant K_D_ ~1 nM	BLItz	Groner et al., 2015 [52]
HIV-1 Gag VLP displaying full-length HER2	HER2-specific phages with Anti-M13-HRP-labelled Ab for signal enhancement and Herceptin-specific mAb	Streptavidin biosensor, biotinylated PEG-Cholesterol anchor on VLPs	PBS; PBST (0.1% Tween-20); PBST + 0.3% skimmed milk powder DAB enhancement in 0.05% DAB-0.015% H_2_O_2_ in PBS	Sf9 expression system is robust for displaying full-length receptors on VLPs. BLI demonstrated functionality of displayed HER2.	Octet QK with eight parallel biosensors	Nika et al., 2017 [53]
Hepatitis B core antigen VLPs with C-terminal His6-peptide, disassembled and intact	Anti-His6 peptide antibody and Hepatitis B VLP-specific antibody	Amino-propylsilane (APS) sensors	Loading with 5% BSA in PBS	Binding of antibodies to assembled or disassembled VLPs elucidates the mechanism of increased stability by the His6-peptide.	Octet RED	Schumacher et al., 2018 [54]

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
