# Peer review of "Applications of Surface Plasmon Resonance and Biolayer Interferometry for Virus–Ligand Binding"

_viruses, 2022, doi:10.3390/v14040717_

Round 1
Reviewer 1 Report
Dear authors, I like your paper. It’s informative and comprehensive. But you said no word about SOI-FET biosensors. There are many publications on this issue, including different viruses’ detection. I think this method should be included in Table 1 and discussed in the text.
The second remark is about paragraph 2 in Introduction. This paper is a review, so this paragraph should contain a lot of references on variety of molecules detection by SPR and BLI. There are none now.
Author Response
Dear Reviewers:
Thank you very much for taking the time to read the manuscript and providing your comments. We have incorporated all three comments into the revised manuscript with track changes enabled:
- We have added references for the different biomolecule interactions in the introduction paragraph 2.
- I was unable to find and access many references on SOI-FET for protein-protein or protein-ligand binding. The paragraph that discusses assays besides SPR and BLI was intentioned to serve as a highlight of some of the more established techniques seen repeatedly in experimental articles. To give the reader the idea that there are more options beyond those in Table 1, I included a brief statement at the end of paragraph 1 in section 1.5 mentioning electrochemical biosensors. I hope this adjustment would be acceptable.
- A real (experimental) full sensorgram has been inserted as Figure 4. We have obtained permission to reproduce the figures from Chenail et al. Analytical Biochemistry (2011) and the figure legend credits the original source. Many of the other data-based sensorgrams in publications are dense and often present several curves at once. To avoid reader confusion, I have included just the one experimental sensorgram. If additional experimental sensorgrams are preferred, we will be happy to request reproduction of more figures and insert into the review.
We hope the revisions are suitable for publication. Thank you and looking forward to hearing from you.
Shruthi Murali
Reviewer 2 Report
The evaluated paper reviews 50 papers concerning virus-ligand binding. Two measuring techniques, SPR and Bilayer Interferometry (BLI), are compared in terms of their effectiveness as tools in the investigation of virus-ligand interaction. A summary of the content of the reviewed papers, such as type of virus, ligand used for the virus immobilisation, method of immobilisation, and instrumental platform is given in four tables. Generally, the review is valuable and skilfully written, and undoubtedly deserves to be published. In my opinion, the review is too synthetic and would be more reader-friendly with two figures showing real sensorgrams: one for SPR and the second for BLI. Therefore, minor revision is recommended.
Author Response

(The authors gave the same response as above.)
